# Non-canonical RNA-DNA differences and other human genomic features are enriched within very short tandem repeats

**Hui Yu**[1]*, **Shilin Zhao**[2], **Scott Ness**[1], **Huining Kang**[1], **Quanhu Sheng**[2], **David C. Samuels**[3], **Olufunmilola Oyebamiji**[1], **Ying-yong Zhao**[4], **Yan Guo**[1]*

**1** Comprehensive Cancer Center, University of New Mexico, Albuquerque, New Mexico, United States of America, **2** Department of Biostatistics, Vanderbilt University Medical Center, Nashville, Tennessee, United States of America, **3** Deptartment of Molecular Physiology and Biophysics, Vanderbilt University, Nashville, Tennessee, United States of America, **4** Key Laboratory of Resource Biology and Biotechnology in Western China, School of Life Sciences, Northwest University, Xi'an, Shaanxi, China

* huiyu1@salud.unm.edu (HY); yaguo@salud.unm.edu (YG)

## Abstract

Very short tandem repeats bear substantial genetic, evolutionary, and pathological significance in genome analyses. Here, we compiled a census of tandem mono-nucleotide/di-nucleotide/tri-nucleotide repeats (MNRs/DNRs/TNRs) in GRCh38, which we term "poly-tracts" in general. Of the human genome, 144.4 million nucleotides (4.7%) are occupied by polytracts, and 0.47 million single nucleotides are identified as polytract hinges, i.e., break-points of tandem polytracts. Preliminary exploration of the census suggested polytract hinge sites and boundaries of AAC polytracts may bear a higher mapping error rate than other polytract regions. Further, we revealed landscapes of polytract enrichment with respect to nearly a hundred genomic features. We found MNRs, DNRs, and TNRs displayed notice-able difference in terms of locational enrichment for miscellaneous genomic features, especially RNA editing events. Non-canonical and C-to-U RNA-editing events are enriched inside and/or adjacent to MNRs, while all categories of RNA-editing events are under-repre-sented in DNRs. A-to-I RNA-editing events are generally under-represented in polytracts. The selective enrichment of non-canonical RNA-editing events within MNR adjacency pro-vides a negative evidence against their authenticity. To enable similar locational enrichment analyses in relation to polytracts, we developed a software Polytrap which can handle 11 ref-erence genomes. Additionally, we compiled polytracts of four model organisms into a Track Hub which can be integrated into USCS Genome Browser as an official track for convenient visualization of polytracts.

## Author summary

Short tandem repeats in the human genome are frequently used as genetic markers in population studies and they are also associated with genetic diseases. Nevertheless, accurate localization and mapping of short tandem repeats in the reference genome have not

**Data Availability Statement:** All computer code of Polytrap are available from the GitHub database (https://github.com/hui-sheen/polytrap/). Genomic coordinates of polytracts in four model organisms

are formatted as a public track for UCSC Genome Browser (http://innovebioinfo.com/Annotation/Polytracts/Polytract.html).

**Funding:** HY, SN, HK, OO, and YG were supported by the cancer center support grant P30CA118100. This study was supported by the Comprehensive Cancer Center at the University of New Mexico, the Bioinformatics Shared Resources and the Biostatistics Shared Resources at The Comprehensive Cancer Center. None of the funding body was involved in the study design, collection, analysis, interpretation of data, or in writing the manuscript.

**Competing interests:** The authors have declared that no competing interests exist.

been well addressed, and there is a lack of systematic catalog of short tandem repeats in the most updated human reference genome. Here, we compiled an updated census of mono-nucleotide/di-nucleotide/tri-nucleotide repeats (MNRs/DNRs/TNRs) from the human reference genome GRCh38, and collectively termed them polytracts. The resultant polytract dataset encompasses TNRs, the presumably more biologically significant repeat species, as well as the under-studied species MNRs and DNRs. With such a composition, the polytract dataset can provide a negative control for genome analyses which are potentially confounded with polytract regions, and it can also be used as a discovery tool to screen for MNR/DNR/TNR-characteristic genomic features. We integrated the polytract dataset with genome coordinates of RNA-editing sites, and found significant enrichment of C-to-U and non-canonical RNA-editing events in adjacency of MNR polytracts, especially break-points of tandem polytracts. The same phenomenon was not observed for the canonical A-to-I RNA editing type. This distinct enrichment patterns between canonical and non-canonical RNA-editing events provides a negative evidence against the authenticity of non-canonical RNA-editing events. Similarly, we examined locations of enhancer sequences relative to polytracts, and found varied degree of locational enrichment among subtypes of polytracts. In practice, different researchers may be interested in different genomic features and/or different organisms, so we developed a tool Polytrap and a Public Track Hub to assist with general locational enrichment analysis of polytracts with respect to localizable genomic features.

## Introduction

In the human genome, short tandem repeats (STRs) of unit size 1–6 bp were estimated to occupy up to two million loci [1] or 3% of the genome [2], and they are believed to bear genetic [3], evolutional [4], and pathological [5] significance. Nevertheless, because their repetitive nature inevitably leads to stutter noise in sequencing [6], STRs are often simply treated as suspicious blacklists [7] in practical Next-Generation Sequencing data analyses. An early catalog of human genome STRs dated back to 2003 [1], at which time the human reference genome was GRCh33. Today, the human reference genome has upgraded to GRCh38, five updates in succession to that antique version. In the past decade, although many efforts were dedicated to calling STR variations from Next-Generation Sequencing data, an updated census of STR in the human reference genome is surprisingly unobserved. As a preparation step to their major analyses, two studies [8, 9] compiled STR catalogs in GRCh37 and GRCh38, respectively; both works employed *ad hoc* procedures and custom parameters, thus generating highly customized datasets unsuitable for general use. In particular, both catalogs were initially generated by an inference-based algorithm TRF [10], so by definition they represented a mixture of perfect and imperfect STRs.

Among all STR species, trinucleotide repeats (TNRs) are the ones most likely to be implicated in human genetic diseases [5], and thus TNRs have unsurprisingly attracted more research efforts [11, 12] than other STR species. Much fewer research efforts have been dedicated to analysis of the shortest repeat species–mono-dinucleotide repeats (MNRs) and di-nucleotide repeats (DNRs). However, a recent discovery that *drosophila* enhancers are characterized by DNRs [13] hints at the biological significance of very short tandem repeats. Additionally, we found that, in the human mitochondrial genome, MNRs or DNRs tend to over-represent tri-allelic heteroplasmy and RNA-DNA differences (RDDs) [14]. The presence of abundant RDDs may result from RNA-editing, a post-transcriptional regulatory mechanism,

but can also represent technical artifacts in both DNA and RNA sequencing. As reviewed earlier [15], RDDs observed in RNA-Seq data comprise real RNA-editing events, single-nucleotide polymorphisms, and artifacts (errors). Actually, there is a strong voice that non-canonical RNA editing events are primarily attributed to sequencing errors or bioinformatics pitfalls [16]. Very short tandem repeats, such as MNRs and DNRs, pose higher sequencing/analysis challenges than other genomic regions, so RDDs around very short tandem repeats may be especially suspicious.

Intrigued by the dispute over non-canonical RDDs [15] and their involvement in mitochondrial MNRs and DNRs [14], we felt a pressing need to expand the analysis of repeat-associated genomic features to the whole contemporary human reference genome (GRCh38), aligning the under-investigated MNRs and DNRs with the more established TNRs. To this end, we first created an up-to-date catalog of precisely defined very short tandem repeats, including MNRs, DNRs, and TNRs, which are termed "polytracts" in general. Different from related works, our polytract catalog was intentionally biased towards very short tandem repeats which comprise MNRs, DNRs, and TNRs, and we included only perfect matches to the polytract definition (see Materials and Methods). After creating the polytract catalog, we revisited prior DNR/TNR-related findings using our updated data. Finally, we performed locational enrichment tests of polytract regions against nearly a hundred genomic features, with an emphasis on RDDs. The present work led to an updated census of MNRs/DNRs/TNRs in human reference genome GRCh38, corroborated and expanded biological significances associated with DNRs and TNRs, and revealed landscapes of genomic features enriched within MNRs/DNRs/TNRs. A software program named Polytrap was developed to assist with locational enrichment analysis of polytracts with respect to localizable genomic features in the human genome (GRCh37 and GRCh38) and other model organisms.

## Results

### A census of MNR/DNR/TNR polytracts in human genome

As detailed in Materials and Methods, we identified MNRs/DNRs/TNRs spanning at least six (for MNR) or three (for DNR and TNR) units in the human reference genome GRCh38, and obtained their summary statistics at genome level (Table 1). MNRs, DNRs, and TNRs occupy 59.1 million (1.9%), 70.6 million (2.3%), and 14.7 million (0.5%) nucleotides, altogether consuming 144.4 million nucleotides (4.7%) of the human reference genome GRCh38 (Fig 1A). Simply comparing the numbers, this latest percentage of 4.7% is much larger than previous estimations of ~3% that were based on early reference genomes [1, 2]. Because the minimum length of STRs was not set identically and the encompassed STR species were not the same, these percentages were not directly comparable to each other. However, we were able to compare these statistics between GRCh38 and GRCh37 in parallel. Using the same protocol, we identified all polytracts in the reference genome GRCh37. Surprisingly, while the total genome size reduced by 0.2% from GRCh37 to GRCh38 [17], the nucleotide volume occupied by MNRs, DNRs, and TNRs consistently increased in absolute number (S1 Table); in total, GRCh37 and GRCh38 harbor 141.6 million (4.6%) and 144.4 million (4.7%) polytract nucleotides, respectively. It seemed that broader regions of polytracts are revealed in newer, more refined reference genomes.

In the field, there is a similar data resource of GRCh38 DNRs/TNRs available as UCSC Microsatellites (http://genome.ucsc.edu/cgi-bin/hgTrackUi?hgsid=104818371&c=chrX&g=microsat). We extracted a comparable subset of DNRs and TNRs from our polytract dataset and compared it with the UCSC Microsatellite dataset. We found that literally 100% of UCSC

**Table 1. Summary statistics of three clades of polytracts in GRCh38.**

| Polytract species | Number of polytracts | Nucleotides enclosed in polytracts | Percentage of polytract nucleotides in genome | Mean length | Median length |
|---|---|---|---|---|---|
| A/T | 7,119,220 | 55,290,931 | 1.79% | 7.8 | 6 |
| C/G | 610,474 | 3,839,875 | 0.12% | 6.3 | 6 |
| *MNR total* | *7,729,694* | *59,130,806* | *1.91%* | *7.6* | *6* |
| TA | 2,764,278 | 19,948,282 | 0.65% | 7.2 | 6 |
| CT/GA | 3,679,922 | 25,030,351 | 0.81% | 6.8 | 6 |
| CA/GT | 3,371,036 | 25,125,048 | 0.81% | 7.5 | 6 |
| GC | 71,160 | 476,554 | 0.02% | 6.7 | 6 |
| *DNR total* | *9,886,396* | *70,580,235* | *2.29%* | *7.1* | *6* |
| AAT | 353,551 | 3,828,512 | 0.12% | 10.8 | 9 |
| ACC | 238,068 | 2,302,369 | 0.07% | 9.7 | 9 |
| AAG | 183,579 | 1,859,914 | 0.06% | 10.1 | 9 |
| AGG | 183,232 | 1,856,718 | 0.06% | 10.1 | 9 |
| AAC | 146,444 | 1,720,479 | 0.06% | 11.7 | 10 |
| CAG | 131,235 | 1,290,213 | 0.04% | 9.8 | 9 |
| ATC | 123,051 | 1,223,622 | 0.04% | 9.9 | 9 |
| ACT | 36,083 | 352,774 | 0.01% | 9.8 | 9 |
| CGG | 21,508 | 238,039 | 0.01% | 11.1 | 10 |
| GAC | 1,396 | 13,915 | 0.00% | 10.0 | 9 |
| *TNR total* | *1,418,147* | *14,686,555* | *0.48%* | *10.4* | *9* |

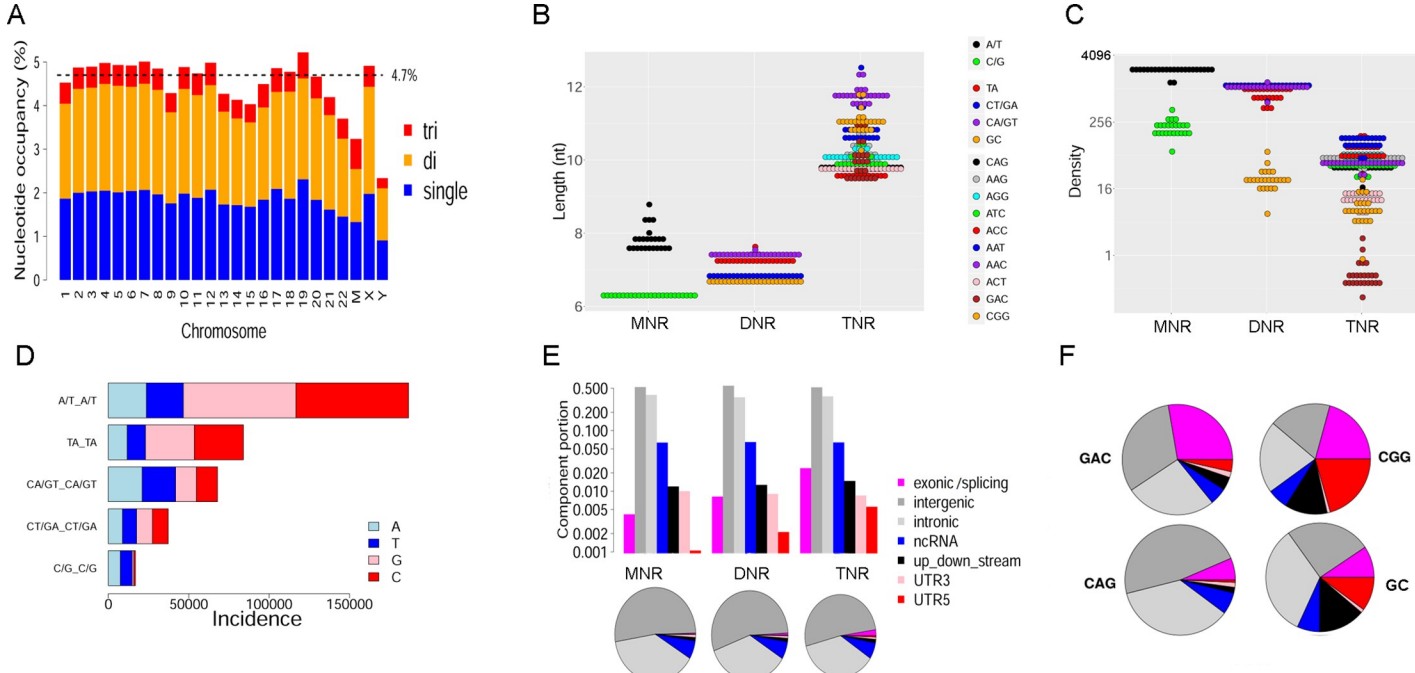

**Fig 1. Landscape of MNR/DNR/TNR polytracts in the human reference genome GRCh38.** A, chromosome-wise polytract occupancy percentages. B, mean polytract length of species of MNR, DNR, or TNR (each chromosome contributes one dot). C, density of each polytract species, i.e., incidence per million base pairs (Mb) along a chromosome. Panel C shares the same color palette with panel B. y-axis is displayed on a log scale. D, quantity and composition of hinges, i.e., single nucleotides connecting tandem polytracts. E, distribution of polytracts over seven genomic regions, displayed in histograms (with log-scaled y-axis) and pie-charts alternatively. Up_down_stream designates the 1-kb genomic segments in proximity to a gene body anchored at transcription start/end site. Exonic/splicing refers to a region that is in an exon or at a splicing junction (intron/exon boundaries). UTR5 and UTR3 denote 5'-UTR and 3'-UTR segments, respectively. F, polytract species with the greatest exome allocation. Panel F shares the same color palette with panel E.

Microsatellites were recovered in the polytract set, but the latter contained additional qualified repeat instances missed by the former (S2 Table).

There is evident disparity in the length of different polytract species (Fig 1B). The mean length of a MNR is 7.6 nt, however the two species of A/T and C/G form two distinct groups (mean 7.8 nt vs. 6.3 nt). The mean length of a DNR is 7.1 nt, but two length groups are visually separable: TA and CA/GT species have longer lengths (mean 7.2~7.5 nt), while CT/GA and GC species have shorter ones (mean 6.7~6.8 nt). The mean length of a TNR is 10.4 nt, with three species showing appreciably longer mean length (AAC 11.7 nt, CGG 11.1 nt, and AAT 10.8 nt).

Even more evident disparity is found with tract density, defined as number of polytracts per Mb (Fig 1C). For MNRs, A/T polytracts are 11 times as dense as C/G polytracts. For DNRs, the densest species is the TA polytract, which is interspersed 41 times as dense as the sparsest species, GC. For TNRs, AAT (density 105/Mb) is the densest species, whereas GAC (0.52/Mb) and CGG (12/Mb) are the sparsest species. A statistical significant linear correlation between mean length and density across chromosomes was found only for TNR tracts but not for MNR or DNR (S1 Fig).

At times, two tandem stretches of polytracts are separated by only one nucleotide, and we term these single-site breakpoints "hinges." There are in total 472,716 hinge sites in GRCh38. As expected, hinge-connected duplexes demonstrate largely the same prevalence order as the polytract unit species *per se* (Fig 1D), with A/T-joining hinges being the most frequent and C/G-joining hinges the rarest.

Based on their location, polytracts were assigned to seven different genomic region categories, and the distribution of each polytract species across the seven region categories was obtained (S2 Fig). Overall, 90.5% of polytracts are located in intronic (36.8%) or intergenic (53.7%) regions, and this quantity does not vary much among the three polytract clades (MNR 90.9%, DNR 90.3%, and TNR 88.1%, respectively; Fig 1E). Notably, TNR species including GAC, CGG, and CAG, as well as the GC DNR, have substantially elevated exonic portion compared to the general pattern (Fig 1F). The same three TNR species were found most over-represented in exome in an early survey [11]. Of note, GAC and CGG are the two least frequent TNR species in the genome, having chromosome density of 0.52/Mb and 12/Mb, respectively (Fig 1C). The CGG TNR and the GC DNR (Fig 1F) also have notable over-representation in up/down-stream segments (1 kb proximity to gene body) and 5'-UTR regions, a phenomenon rarely seen in other polytract species (S2 Fig).

## Corroborated and extended biological relevance of DNR and TNR

Previously, a study of thousands of enhancer sequences in three *Drosophila* cell lines proposed that DNRs are a general enhancer feature, especially with respect to universal enhancers [13]. Herein, of all 5,967 human enhancers, we found 4,677 (78.4%) embedded or adjoined at least one polytract. Each individual clade, MNR, DNR, and TNR, as well as the whole polytract set, were significantly embedded in enhancers (Bonferroni-adjusted $p < 0.01$, hypergeometric test; Fig 2A). As shown in Fig 2B, of MNRs, C/G polytracts were more likely to be embedded in enhancers than A/T polytracts (Relative Risk (RR): 84.6 vs. 11.0); of DNRs, GC polytracts were most likely to be embedded in enhancers with a striking RR of 173.6; of TNRs, CGG, AGG, and CAG polytracts stood out ahead of other species, showing RRs 173.0, 85.4, and 68.9, respectively. The TA DNR showed the weakest trend of being embedded in enhancers among the four DNR species (RR = 4.9), which is accordant with the observation in *Drosophila* [13].

Compared with MNRs, DNRs were more likely to be embedded in enhancer sequences (RR 22.8 vs. 16.0, Fig 2A), and the embedding tendency in DNR was stronger for universal enhancers than specific enhancers (Fig 2C). These observations were again coherent with findings in

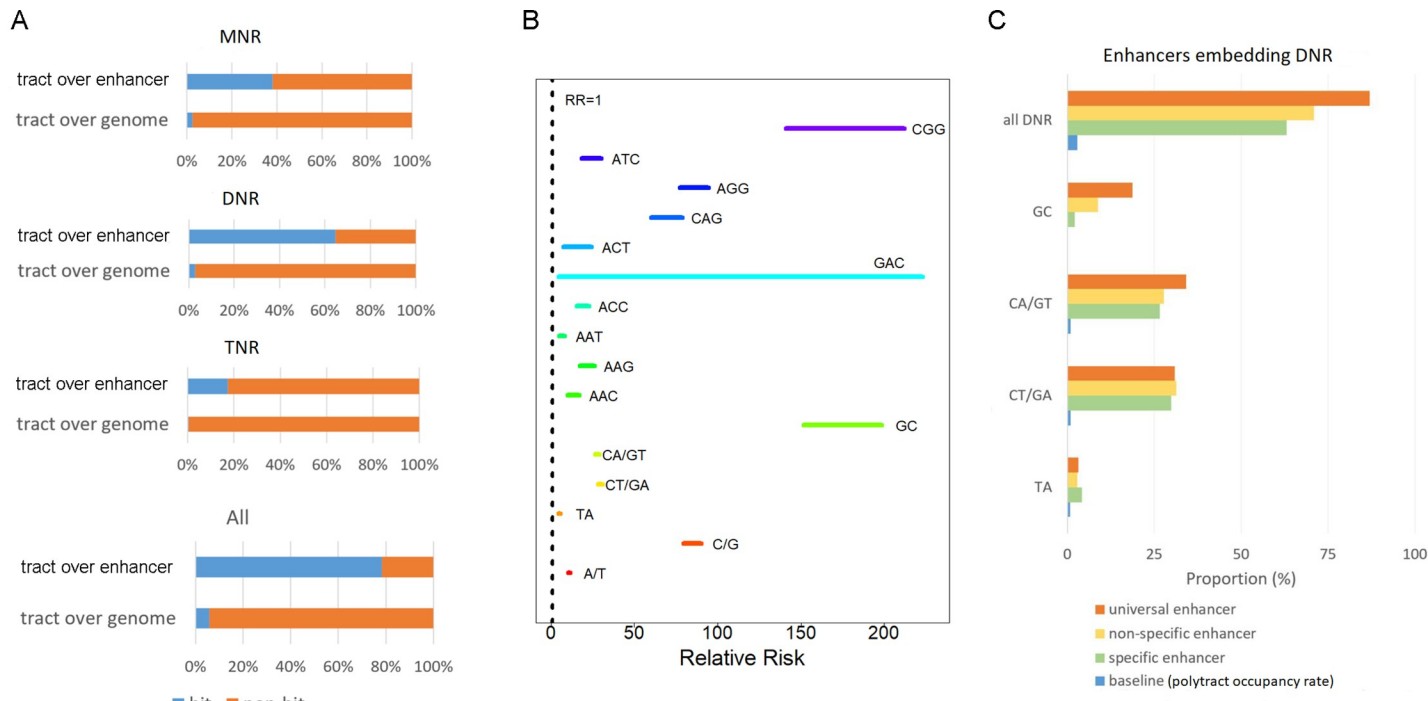

**Fig 2. Enhancers are significantly enriched with polytracts.** A, proportion of enhancers embedding polytracts ("tract over enhancer") largely surpasses the baseline proportion ("tract over genome"), obtained through dividing polytract nucleotide quantity with the total size of the reference genome. B, polytracts' 95% confidence interval of Relative Risk of being embedded in enhancers. C, association rate for universal/specific enhancers with DNR. As a superset to "universal enhancer," "non-specific enhancer" is complementary to "specific enhancer." Baseline is included as the occupancy percentages of each DNR species in GRCh38.

the *Drosophila* study [13]. Moreover, we noted an even stronger tendency for enhancers to embed TNRs (RRs 30.9, Fig 2B). To our best knowledge, this is the first time that TNR is identified as a feature of enhancer sequences, especially in the human genome.

Previously, a systematic genomic investigation [11] was conducted on TNRs in the human genome (GRCh36), where the minimum repeat length was set at 18 nt (or "6U" for 6 repeat units). To align our data with the precedent design [11], from the total 1,418,147 TNR tracts, we identified a subset of 39,937 long polytracts that each contained six or more (complete or incomplete) trimer units (i.e., length $\geq$ 18nt). By repeating the same precedent analytics [11], we revealed largely the same length distributions for the ten TNR species, and rendered a similar TNR clustering based on a matrix of pairwise Kolmogorov-Smirnov test statistics (S3 Fig).

In addition to long repeat tracts of 6 units or more [11], our full TNR polytract set contains a dominant portion (97.2%) of five-unit long or less, which had not been touched on yet. We repeated the same analysis on this tremendously larger, full TNR set, discriminating difference in tract length distributions among TNR species. Interestingly, this moderate-repeat-dominated dataset discriminated two TNR types inconspicuous in the prior study, namely CGG and AAC (Fig 3A and 3B). It is visually discernable that these two TNR species have the highest portions of long repeats (6U or more) than do all other species (Fig 3A). AAC has the most dissimilar length distribution, with the greatest portion (61.8%) of tracts enclosing ten or more nucleotides; no wonder it has the longest average length (Fig 1B). Another peculiar phenomenon with AAC is that the polytract frequency does not decrease monotonously with increasing polytract length; two surprising modes appear at 11 nt and 14 nt, the lengths awaiting a single nucleotide to form intact TNR cycles. Investigating further, we found pre-cycle length modes

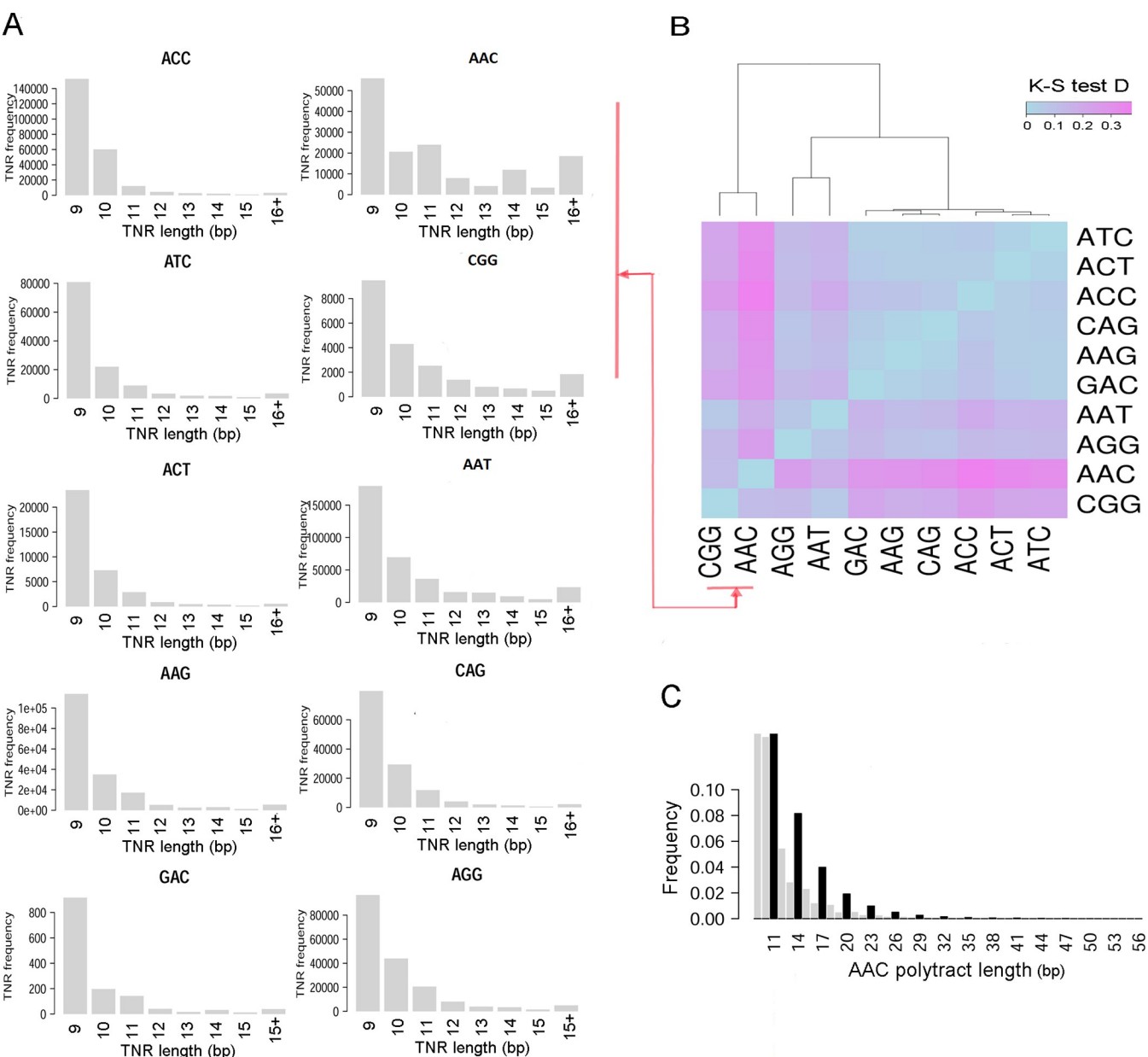

**Fig 3. Length distributions of Trinucleotide repeats (TNR) in the human genome.** A, polytract length distribution dissected by TNR species. B, heatmap graph showing the Kolmogorov-Smirnov statistics *D* resulting from pairwise comparisons of polytract lengths among all 10 TNR subtypes. Hierarchical dendrogram highlights CGG and AAC as unique from the rest majority. Distance metric used one minus correlation value and linkage choice was the average method. C, full-range length distribution of AAC polytracts. Dark black bars highlight periodical pre-cycle modes at length 11 bp, 14 bp, 17 bp, etc..

repetitively appear in a majority part (3~17 U) of the whole length range (3~19 U) of AAC tracts (Fig 3C).

## Non-canonical RNA-editing events are enriched within MNR adjacency or hinge sites

We obtained genome locations for 4,688,495 RNA editing events, which were categorized to three classes: A-to-I, C-to-U, and non-canonical. Because in Next-Generation Sequencing

**Table 2. RNA editing event classes.**

| Practical class name | Standard class name | Nucleotide changes | Incidence |
|---|---|---|---|
| A-to-G | A-to-I | A>G, T>C | 4,677,846 |
| C-to-T | C-to-U | C>T, G>A | 5,006 |
| non-canonical | non-canonical | All changes except A-to-G and C-to-T | 5,643 |

experiments inosine (I) and uridine (U) are replaced with guanine (G) and thymine (T), respectively, we renamed the two canonical classes as A-to-G and C-to-T, respectively (Table 2). Through the binomial probability model (details in Materials and Methods), we examined the locational enrichment tendency of various RNA-editing classes in various polytract clades or species. In a nutshell, we evaluated if RNA editing events were over-represented in the polytract territory as compared to the baseline frequency across the whole genome. Here, the polytract territory referred to the aggregate segments of the genome occupied by all polytracts, and it was their collective nucleotide volume that we quantified. Notably, in addition to the exact territory of polytracts, we extended each polytract one nucleotide bi-directionally and thus formed an adjacency-extended polytract territory. Locational enrichment analysis was performed in parallel between the exact polytract territory and the extended polytract territory.

Of all three polytract clades, MNR displayed the most disparate enrichment pattern across RNA editing classes. In terms of the exact polytract territory, MNR enriched non-canonical editing events (Binomial p<1e-22) only, but in terms of the extended polytract territory, MNR enriched both C-to-T events and non-canonical events (Fig 4A, "MNR" panel). All classes of

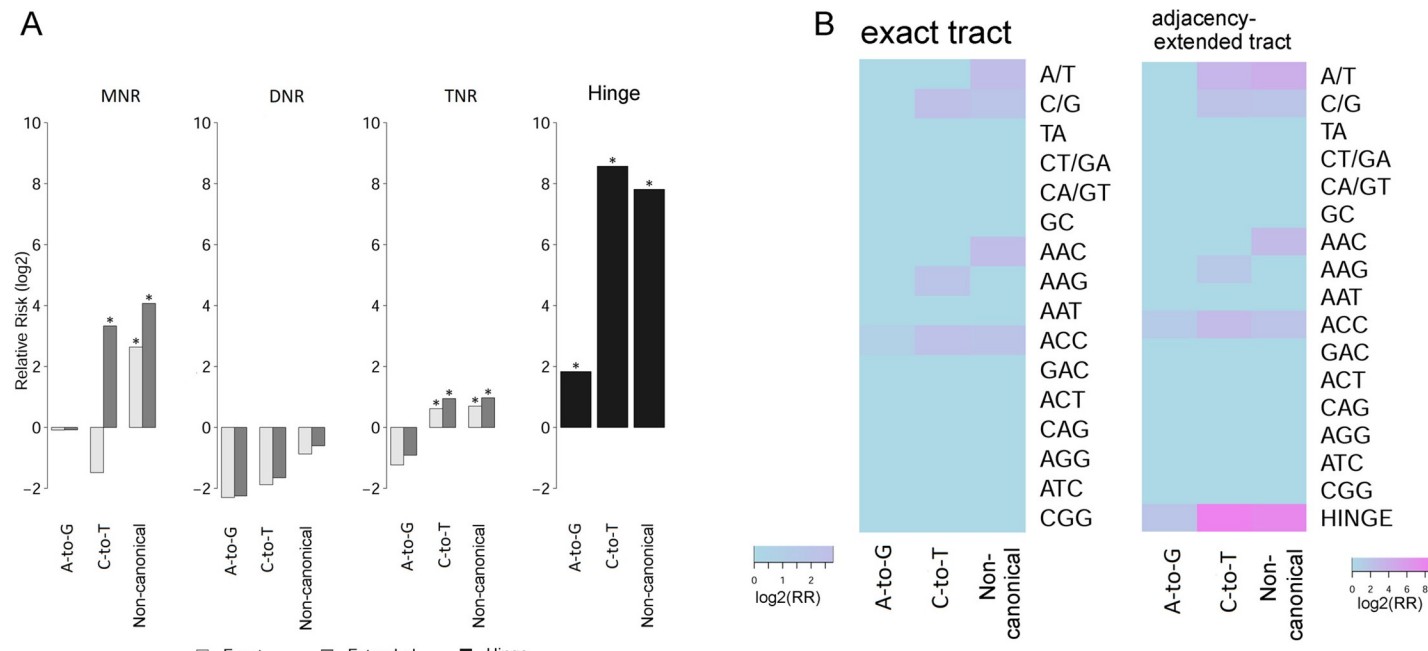

**Fig 4. Non-canonical RNA-DNA differences are enriched within polytracts and, more markedly, break-points (hinges).** A, A-to-G and C-to-T editing events do not have a higher likelihood of appearing in MNRs, whereas non-canonical events do; A-to-G, C-to-T, and non-canonical RNA-editing events all have a higher likelihood of appearing in polytract hinges. *Exact*, exact regions of polytract occupancy. *Extended*, polytracts extended with one neighboring nucleotide bi-directionally. *Hinge*, single-nucleotide sites located between two neighboring stretches of exact polytracts. Asterisk (*), binomial enrichment p<0.01. B, heatmap of Relative Risk (RR) of polytract bearing an RNA-editing event, depicted for combinations between editing event classes and polytract species. Non-elevated relative risks (RR<1) or statistically insignificant (p>0.01) associations of editing events were imputed with RR = 1.

RNA-editing events, including A-to-G, C-to-T, and non-canonical, showed elevated RR of residing in hinge sites (Binomial test p<1e-22 for all; Fig 4A, "Hinge" panel), and the RRs with C-to-T (RR = 380) and non-canonical (RR = 225) were much higher than A-to-G (RR = 3.6). Tandem duplexes of A/T MNRs accounted for a predominant portion of these RNA-editing–concurrent hinge sites—90.4% for A-to-G, 99.3% for C-to-T, and 96.2% for non-canonical. It might be that an actual longer, intact A/T fragment is incorrectly broken into two pieces in the reference genome, which leads to spurious reports of C-to-T RDDs in hinges of A/T MNRs.

Considering the nucleotide substitution source and target in C-to-T editing events, we may regard C/G MNRs as the "source" polytracts and A/T MNRs as the "target" tracts. Therefore, it is expected to see C-to-T events condense in the source, C/G MNR (Fig 4B,"exact tract" panel). With bi-directional 1-nt extension, however, C-to-T events demonstrate significant over-representation in both the source, C/G, and the target, A/T (Fig 4B, "adjacency-extended tract" panel). The spurious enrichment of C-to-T events in adjacency of A/T polytracts may also be explained if we presume adjacency of A/T polytracts bears more mapping errors than the reference genome average. While the presumption of polytract hinges and adjacencies bearing more mapping errors is intuitively plausible, wet-lab experiments are needed to verify these suppositions and possibly to discriminate false C-to-T editing events.

Overall, A-to-G events did not show evident locational preference of any tract species, except for a weak enrichment tendency within polytract hinge sites (RR = 3.6). None of the three classes of RNA-editing events was found enriched within DNRs (Fig 4A and 4B). Occasional locational enrichment took place between certain editing classes and TNR species (Fig 4B). Notably, the AAC polytract, a moderately prevalent TNR species with distinctive pre-cycle length modes, was significantly enriched with non-canonical editing events only (RR = 7.1, binomial test p = 4.2e-5, Fig 4B). We speculate that many non-fully-cycled AAC tracts have wrong trailing single nucleotides, which led to the counter-intuition pre-cycle length modes (Fig 3C) and clustering of spurious non-canonical RNA-editing events (Fig 4B). In future refinement of human reference genome, special attention should be given to suspicious trailing sites to incomplete AAC TNRs.

## Landscape of genomic feature locational enrichment within polytracts

Moving beyond enhancers and RNA-editing sites, we interrogated polytract locational enrichment with respect to miscellaneous genomic features, including categorized gene regions, RNA-binding protein's presumable binding segments [18], polymorphism variants [19], somatic mutations [20], etc., thereby resulting in landscapes of polytract locational enrichment against 94 individual genomic features (Fig 5, S3 Table). Globally, TNRs are associated with the largest number of genomic features, and DNRs the least. MNR-enriched features include non-canonical RNA-editing events, enhancers, non-coding RNAs, polymorphism variants, eQTLs, histone modification sites, retrotransposons, and binding segments of several RNA-binding proteins. Different from MNRs, TNRs show over-representation for certain genes (protein_coding genes, pseudogenes, and immunoglobulin genes), but not for any kind of RNA-editing events. DNRs are not associated with either RNA-editing events or gene regions; a specific small RNA ("sRNA") stands out only because of the dimer-rich paralogs of *Clostridiales-1 RNA (RF01699)*.

We also noted several interesting phenomena in relation to specific genomic features. Exome analysis blacklist variants are enriched within all three clades of polytracts, and the statistical significance indicates such an order: MNR (p = 2.2e-217) > DNR (p = 1.6e-106) > TNR (p = 7.3e-28). Transcription factors ("TFBS") have a tendency to bind into MNR, DNR, and TNR polytracts, whereas most RNA-binding proteins do not, except for selected TNR-

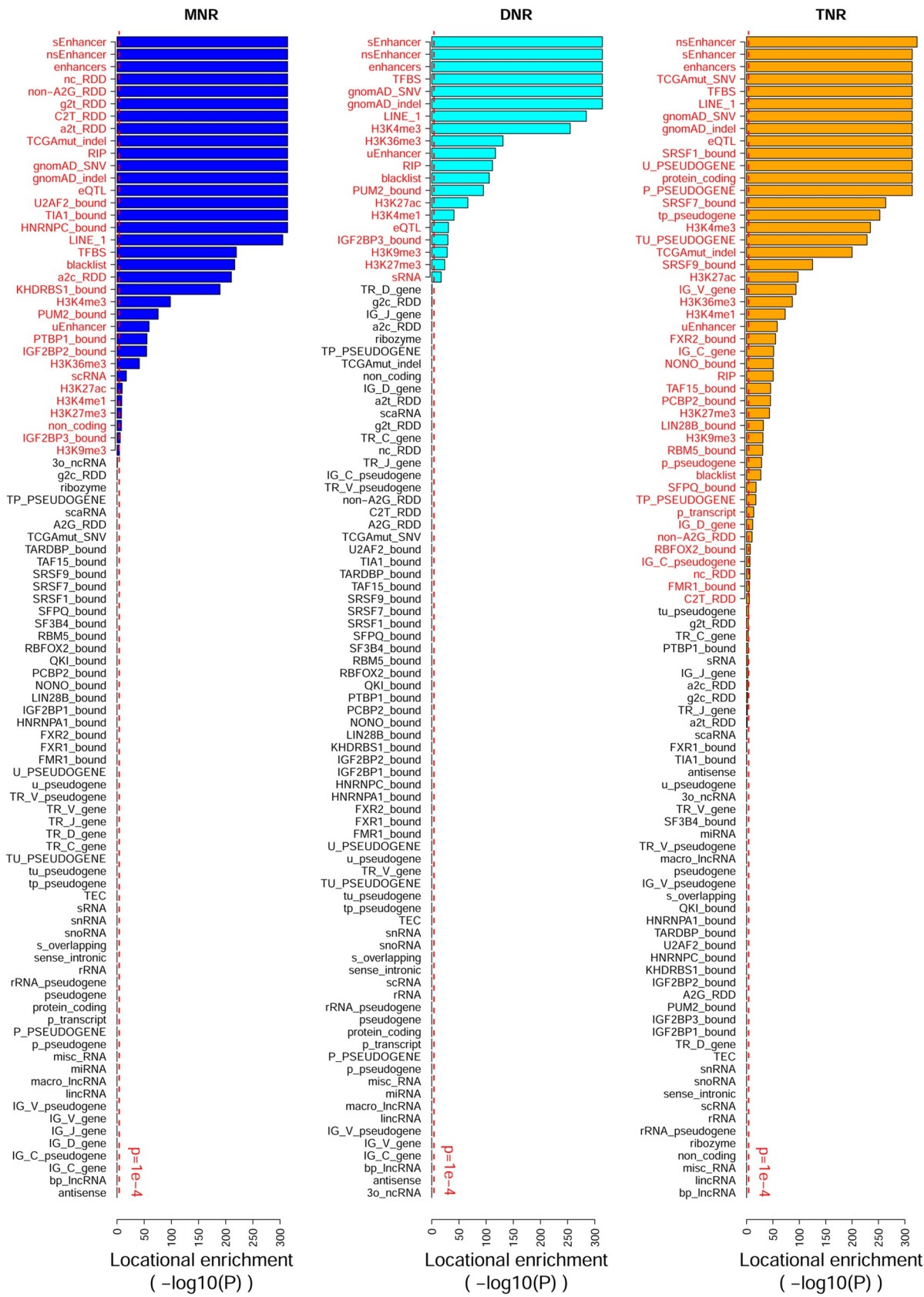

**Fig 5. Landscape of genomic feature enrichment in MNR, DNR, and TNR polytracts.** Features manifested in red are deemed statistically significantly enriched. Because nearly 100 features were tested simultaneously, we applied bonferroni correction and deemed p<1e-4 as statistically significant.

favoring ones, especially splicing factors. Polymorphism variants, including single nucleotide variation ("gnomAD_SNV") and indels ("gnomAD_indel"), tend to cluster around any clade of polytracts, whereas cancer somatic mutations are only over-represented in MNRs and TNRs. Within somatic mutations, whereas indels are over-represented in both MNRs and TNRs, SNVs (single nucleotide variations) are enriched in TNRs only. Because our somatic mutation data were severely biased towards exome and certain TNR species were over-represented in exome, we were afraid the exclusive enrichment of somatic SNVs in TNR was merely a consequence of these two known biases. So we identified the subset of non-coding somatic SNVs from the total somatic SNVs and performed locational enrichment analysis on them only. Still, non-coding somatic SNVs displayed locational enrichment in TNRs only (S4 Fig).

## A software package to evaluate genomic feature enrichment within polytracts

To assist with general locational enrichment analysis with respect to any custom genomic features, we developed a Linux package named Polytrap. With input genomic intervals specified as start and end coordinates on chromosomes, Polytrap distinguishes intervals overlapping with any instance of polytracts, and assesses statistical significance of interval enrichment in polytract clades and species (Fig 6). Our bulk locational enrichment analysis against 94 individual genomic features, as expounded above, was assisted by Polytrap. We recorded the time and memory usage pertinent to RNA-editing analyses (Table 3) to enable estimation of practical computational costs, which is proportional to the size of the input dataset.

While some existent genomic analysis tools bear partial capabilities in relevance to STR, Polytrap appears as the only tool to identify and assess the locational enrichment of user-defined genomic intervals within very short tandem repeats in a systematical and straightforward manner (A theoretical comparison is provided in S4 Table). Polytrap allows to focus on subsets of polytracts with or without adjacency extension, or polytracts embedded by particular gene body regions (protein-coding genes, lncRNAs, or pseudogenes). While all analyses presented above were testing over-representation of a genomic feature, we ensured Polytrap have the flexibility of testing under-representation as well. Taking advantage of this option, we investigated the under-representation tendency of different classes of RNA-editing events in polytracts, and found A-to-G events were under-represented in almost all polytract species, excluding A/T, ACC, and hinges (S5 Fig).

Beyond the human genome, STRs receive increasing attention in non-human organisms as well, including macaques [21], dogs [22], *fugu* [23], and plants [24]. To maximize the potential of Polytrap, we made it capable of handling nine organisms (human, macaque, mouse, rat, dog, chicken, zebrafish, fruitfly, and yeast) and allowed it extendable to uncovered organisms with provisions of Bioconductor's genome support (DOI: 10.18129/B9.bioc.BSgenome). Polytract data files for four organisms (human, mouse, rat, and fruitfly) are formatted into a Track Data Hub [25] for convenient visualization at UCSC Genome Browser (http://genome.ucsc.edu/). The software package and related data files are released on GitHub (https://github.com/hui-sheen/polytrap/) and our project webpage (http://innovebioinfo.com/Annotation/Polytracts/Polytract.html).

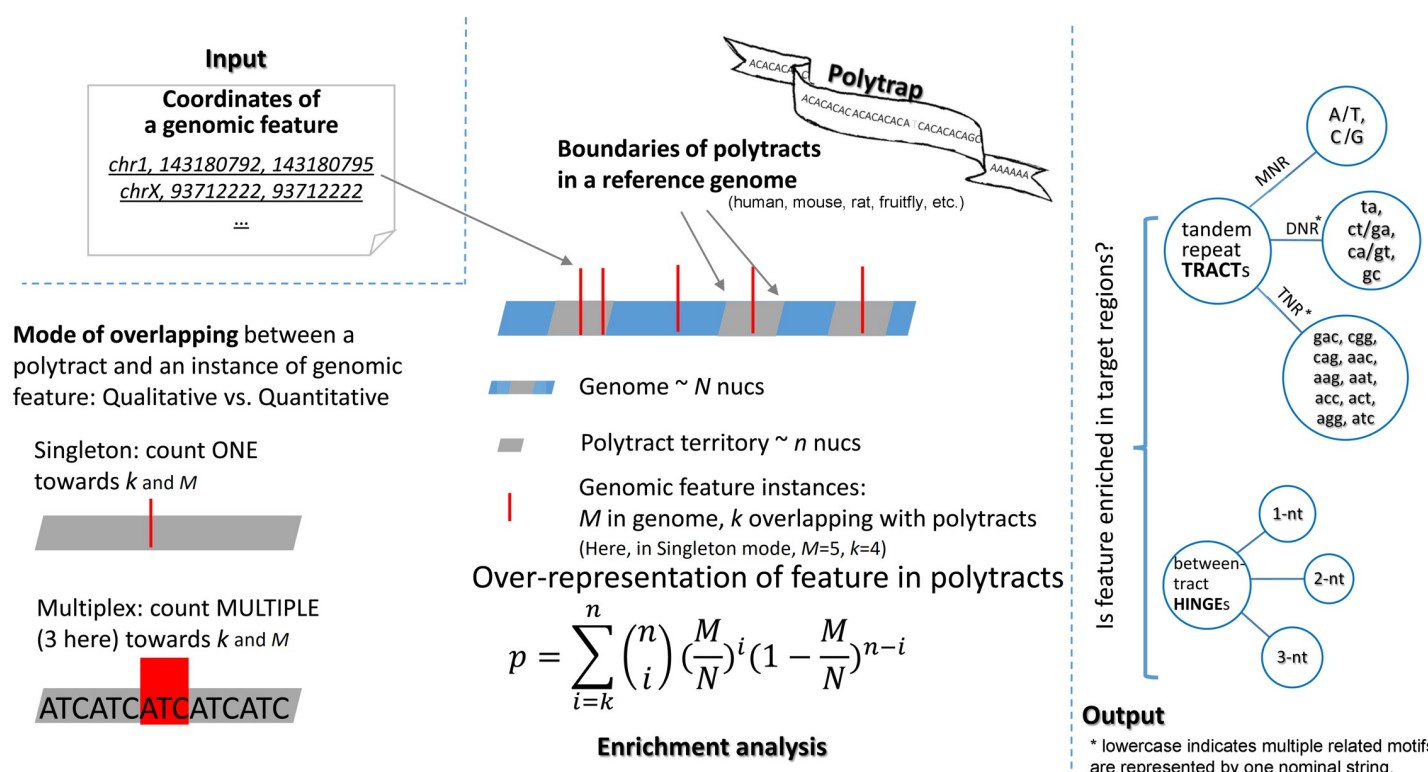

**Fig 6. Schema of software package Polytrap.** Polytrap stores chromosome locations of polytracts (1-mer, 2-mer, and 3-mer tandem repeats, termed MNR, DNR, and TNR) in a reference genome, and calculates the probability of enrichment of user-given genomic feature within polytracts. For a genome of totally *N* nucleotides and a polytract territory of *n* nucleotides, the probability of *k* out of *M* instances of a genomic feature being located within tracts is calculated through the binomial model. When the overlapping mode is switched from "singleton" to "multiplex," *k* and *M* designate number of aggregate nucleotides instead of instances (bottom left part).

## Discussion

STRs are receiving increasing appreciation for their roles as genetic markers, polymorphism variants, origins of heritable neurological diseases [3]. Recently, several analysis tools were developed to facilitate STR variant calling from sequencing data [6, 26, 27]. Among the increasing research efforts, we noted a lack of systematic census and characterization of STRs in the updated human reference genome. A careful scrutiny and characterization of STRs in the reference genome will benefit the expanding STR-focused personal genome analyses. In

**Table 3. Polytrap computational cost on example sessions.** Sessions were tested on a Linux Ubuntu work station with Intel Xeon CUP E5-2650 V4 @ 2.20GHz and 32 GB memory.

| Analysis modality | Input dataset size* | Time usage | Memory usage |
|---|---|---|---|
| TNR | 5K | 0m16s | 1.7G |
|  | 5M | 1m14s |  |
| DNR+TNR | 5K | 1m43s | 7.6G |
|  | 5M | 2m49s |  |
| MNR+DNR+TNR | 5K | 3m24s | 14.2G |
|  | 5M | 4m35s |  |

*dataset size refers to the number of nucleotide positions involved in the input data file. In these example sessions, 5K corresponds to our C-to-T dataset (5,005 positions) and 5M corresponds to our A-to-G dataset (4.7 million positions).

this work, we found 4.7% of the human genome (GRCh38) is occupied by polytracts, and there are 0.47 million single nucleotides located between tandem polytracts, which we call polytract hinges. More than a half of these hinge sites break apart thymine-involved polytracts. Unlike other TNR species, the AAC polytract shows suspicious pre-cycle modes in length distribution. It is a commonsense that STR boundaries bear a higher mapping error rate than the genome baseline, and our results might further pinpoint the hinge sites and AAC polytract boundaries may bear an exceptionally higher mapping error rate.

In accordance with its inherent error-prone nature, polytracts demonstrate enrichment of non-canonical RNA-editing events, somatic indels, and blacklist variants. Thus, polytracts can reasonably serve as a negative reference set for quality controlling genome/exome sequencing studies, helping to flag suspicious variant calls or other sequencing results that are significantly biased to polytract regions. For this negative quality control purpose, we suggest more weights be given to MNR than DNR and TNR, because monomer runs are especially intractable regions in sequencing experiments thus a lower data quality may be associated with MNR in terms of both the mapped sequence in the reference genome and the re-mapped sequence in a personal genome. Across the comparative enrichment landscapes of MNR/DNR/TNR, somatic SNVs are enriched in TNR only whereas somatic indels are enriched in all three polytract clades. In practical sequencing projects, downstream analyses in succession to variant calling usually trust SNVs more than indels, and a component of too many indels raises a warning sign. The disparate polytract enrichment pattern between SNV and indel is consistent with our varied confidence towards SNV and indel in sequencing practices. Likewise, the blacklist variants display the highest statistical significance of enrichment in MNR yet the lowest statistical significance in TNR, again discriminating the more error-prone nature of MNR. So, it is advisable that sequencing practitioners include polytract/MNR enrichment analysis in their quality control protocols.

Polytracts comprise DNRs and TNRs in addition to MNRs. TNRs are appreciated more as biologically significant motifs than avoidable pitfalls. In the polytract enrichment landscapes, TNR does show significant enrichment for many meaningful cis-elements, including transcription-factor binding sites and RNA-binding protein target segments. Interestingly, many genomic features are enriched in MNR/DNR as well as in TNR. Especially, DNR resembles TNR in demonstrating strong over-representation of enhancers and strong under-representation of A-to-I RNA-editing events, thus indicative of remarkable biological significance. MNR also shows significant enrichment for some meaningful genomic features, including binding targets for select RNA-binding proteins. RNA-binding proteins HNRNPC, TIA1, and U2AF2 all favor U-rich binding motifs and their target sequences show distinctively lower entropy than common RNA-binding proteins [28]; concordantly, binding segments of these three RNA-binding proteins enrich MNR yet not DNR or TNR (Fig 5). In this context, the enrichment within MNR indicates differential characteristics of binding motifs of select RNA-binding proteins. Therefore, it risks neglecting genuine biological signals if we always ignore or downweight features/entities that are enriched in simple repeats such as MNRs. Generally speaking, domain knowledge and meticulous inspection are helpful for interpreting polytract enrichment results rationally.

RNA-editing events are classified as canonical events and non-canonical events, and A-to-I events dominate over C-to-U events in the canonical category. There has been a long-lasting dispute over the authenticity of non-canonical editing events [15]. Peer researchers pointed out flaws in designs or protocols of studies that advocate the universality of non-canonical editing events [29, 30]. Here, we discovered that non-canonical RNA-DNA differences are overrepresented in MNR adjacent sites or hinge sites whereas A-to-I events are generally under-represented in polytracts. From a novel angle, the selective enrichment of non-canonical RNA-editing events within MNR adjacency provides a negative evidence against their authenticity, and it also supports the practical operation of assessing the quality of an RDD

dataset by the proportion of A-to-I events [31]. We even argue that a considerable portion of C-to-T editing events are false discoveries, as C-to-T editing events are condensed in the vicinity and hinge sites of polytracts, resembling non-canonical events but not A-to-I events. Our suspicion on C-to-T editing events might be linked to a recent finding that human C-to-U coding RNA editing is largely nonadaptive and that they probably manifest cellular errors [32]. The predominant canonical editing class, A-to-I, is under-represented in all polytract species except A/T MNR, ACC TNR, and hinges. Again, it is alerted that special caution should be cast on polytract hinge sites and boundary regions of certain repeat types.

STRs constitute an abundant component of genomic DNA in not only humans but also many other species, such as macaques [21], *fugu* [23], and plants [24]. Our work developed a convenient tool, Polytrap, to aid with locational enrichment analysis with respect to polytracts in the reference genomes of diverse model organisms. Polytrap can be used as a quality control tool for genome analyses which are potentially confounded with polytract regions, or as a discovery tool to screen for particular MNR/DNR/TNR polytracts characterized with the user-interested genomic features.

## Conclusions

Our work resulted in a comprehensive census of MNR, DNR, and TNR in the latest human reference genome (GRCh38) and a software Polytrap to assist with general locational enrichment analysis of polytracts with respect to localizable genomic features. We found a plethora of genomic features are significantly co-localized with polytracts, including both meaningful biological signals (enhancers, transcription factor binding sites, etc.) and artifact-beset entities (insertions/deletions, blacklist variants, etc.). Most notably, non-canonical RNA editing events are enriched inside and/or adjacent to MNRs, whereas A-to-I editing events are generally under-represented in polytracts. This provides a negative evidence against the authenticity of non-canonical RNA editing events. Polytracts can be referenced as a negative quality control for sequencing studies, but they also bear the potential to inform genomic researches. Polytrap enables swift locational enrichment analysis of subtyped polytracts, and we encourage knowledgeable and meticulous inspection in interpreting the polytract enrichment results.

## Materials and methods

### Identification and categorization of polytracts

In a reference genome, a polytract is defined as a tract of mono-nucleotide, di-nucleotide, or tri-nucleotide tandemly repeated motifs, with possibly incomplete terminal motif included, where the minimum number of repeated units are 6 for MNR and 3 for DNR and TNR. Unlike many related works which allow fuzzy STRs, we seek only perfect matches to our definition. This was done through a string matching between a pattern and each chromosome sequence, with assistance from R packages "stringr" (https://CRAN.R-project.org/package=stringr) and "BSgenome.Hsapiens.UCSC.hg38" (www.bioconductor.org; DOI: 10.18129/B9.bioc. BSgenome). Because of the complementarity between a purine and pyrimidine pair (A:T and G:C), we combined poly-A and poly-T runs into an "A/T" group, poly-G and poly-C runs into a "G/C" group, poly-CT and poly-GA runs into a "CT/GA" group, and poly-CA and poly-GT runs into a "CA/GT" group. Similarly, a nominal full set of 60 TNR motifs were merged into 10 groups: **AAC** (accommodating AAC, ACA, CAA, GTT, TGT, and TTG), **AAG** (AAG, AGA, GAA, CTT, TCT, and TTC), **AAT** (AAT, ATA, TAA, ATT, TAT, TTA), **ACC** (ACC, CCA, CAC, GGT, TGG, GTG), **GAC** (GAC, ACG, CGA, GTC, CGT, and TCG), **ACT** (ACT, CTA, TAC, AGT, TAG, and GTA), **CAG** (CAG, AGC, GCA, CTG, GCT, and TGC), **AGG** (AGG, GGA, GAG, CCT, TCC, and CTC), **ATC** (ATC, TCA, CAT, GAT, TGA, and ATG),

and **CGG** (CGG, GGC, GCG, CCG, GCC, and CGC). In summary, our resultant polytract dataset consisted of two types of MNRs, four types of DNRs, and ten types of TNRs (Table 1). All instances of GRCh38 polytracts are stored as data files in our public software Polytrap (https://github.com/hui-sheen/polytrap/ and http://innovebioinfo.com/Annotation/Polytracts/Polytract.html).

In the same manner as we dealt with the reference genome GRCh38, we identified all polytracts in the reference genome GRCh37. In our miscellaneous analyses, certain data resources (including enhancers and RNA editing sites) were based on GRCh37 instead of GRCh38, and in such cases we performed the analyses against GRCh37 polytracts.

## Locational enrichment analysis of polytracts with respect to localizable genomic features

Having located each polytract in the reference genome, we sought to evaluate the degree of enrichment of a particular genomic feature within the polytract territory, where the genomic feature is represented as a set of genomic sites or intervals and the polytract territory refers to the aggregate segments of the genome occupied by all polytracts (Fig 6). This is a general question of assessing the (unexpected) locational overlapping between one set of genomic intervals and another set of genomic intervals, and previous studies [33–35] have provided solutions of a largely same principle. Basically, we define the expected frequency of observing the genomic feature within the polytract territory as $n/N$, where N denotes the total size of the genome and n the size of the polytract territory; when $k$ out of $M$ instances of the genomic feature fall into the polytract territory, if the observed frequency $k/M$ is much higher than $n/N$, it indicates probable enrichment. Classical probability models such as hypergeometric model and binomial model are frequently used in such scenarios to test for locational enrichment. Here, we followed early examples [33, 35] to employ the binomial model.

*H0 (null hypothesis)—the concerned genomic feature is distributed in polytract territory at a frequency no higher than random expectation.*

*H1 (alternative hypothesis)—the concerned genomic feature is distributed in polytract territory at a frequency higher than random expectation.*

Central to the hypothesis test was identification and quantification of overlaps between polytracts and instances of the genomic feature. We conceived of two modes for this issue. Under the default "singleton" mode, we regarded a genomic interval as a "singleton" unit, so whenever a spatial overlap appeared between a tract and an instance of the genomic feature, we counted it as ONE overlap. Under the alternative "multiplex" mode, we regarded a genomic interval as a union of its constituent nucleotides, so, when a spatial overlap appeared, we counted MULTIPLE nucleotides towards the overlap quantity (Fig 6).

Under the null hypothesis and the singleton mode, for a genome of totally $N$ nucleotides and a tract territory of $n$ nucleotides, the probability of $k$ out of $M$ instances of a genomic feature being located within tracts was calculated through the binomial model (Eq 1). The Relative Risk of polytract bearing the concerned genomic feature was obtained by dividing the tract-ridden rate by the tract occupancy rate (Eq 2).

$$p = \sum_{i=k}^{n} \binom{n}{i} \left(\frac{M}{N}\right)^i \left(1 - \frac{M}{N}\right)^{n-i} \qquad \text{Eq1}$$

$$RR = \frac{k/M}{n/N} \qquad \text{Eq2}$$

When we interrogated polytract association with various types of gene regions and RNA-binding protein target segments, we imposed the multiplex overlap mode. There, the probability of null hypothesis was calculated using Eq 1 too, however in such cases $M$ referred to the total nucleotide quantity of the genomic feature and $k$ denoted the quantity of overlaid nucleotides in all overlapping instances.

By default, Polytrap conducts a one-tailed test through the binomial distribution model, testing for the over-representation tendency of a genomic feature within the polytract territory (Eq 1). However, we ensured Polytrap have the flexibility of testing for the under-representation tendency as well. By setting the directionality option to under-representation, Polytrap calculates the probability of reaching no more than the observed locational overlapping instances (Eq 3), thus effectively assesses the under-representation tendency of a genomic feature within the polytract territory.

$$p = \sum_{i=0}^{k} \binom{n}{i} \left(\frac{M}{N}\right)^{i} \left(1 - \frac{M}{N}\right)^{n-i} \qquad \text{Eq3}$$

Lastly, it is worth noting that, in assessment of feature enrichment within polytract territory, we extended each polytract instance to the single immediate neighboring nucleotide bi-directionally. This operation was inherited from our previous related study [14] and it was set as a default but suppressible choice in Polytrap.

## Genomic features enrolled in the locational enrichment analysis

Having identified the polytracts in GRCh38 and GRCh37, respectively, we went on to investigate possible locational enrichment tendency of polytracts with respect to an array of genomic features. Ninety-four genomic features localized in GRCh38 or GRCh37 were coarsely grouped to five major classes, namely RDD, enhancer, Gene region in HG38, RNA-binding protein's binding segments, and miscellaneous (S3 Table). We recruited RDD foremost because RDDs displayed an intriguing enrichment in mitochondrial polytracts [14] and we wanted to verify if the same phenomenon exists in the nuclear genome. Enhancer was chosen because an early study discovered characterization of *drosophila* enhancer with DNR [13] and we wanted to verify if the same phenomenon exists in the human genome. Diverse gene regions in HG38 were separately analyzed to yield a comparative view of polytract distribution across different types of gene regions. Certain RNA-binding proteins' binding motifs manifest inherent nucleotide repeats, so the imminent analysis would be able to confirm such internal repeat feature and also possibly indicate nearby or distant repeat features. Finally, more than ten miscellaneous features were included the analysis because they were sequential features with easily attainable genomic locations.

RNA editing events (GRCh37) were merged from REDIportal [36] and DARNED [37], reaching a total of ~4.67m. Because the source databases reported editing events on forward strand and strand indistinguishably (e.g., both A>G and T>C denote the A-to-G class), we grouped all editing events into six classes, which further formed three major categories: A-to-G, C-to-T, and non-canonical (Table 2).

Genomic coordinates of 5,967 enhancers in the human genome GRCh37, originally from FANTOM5, were obtained from a published supporting material [38]. These enhancers were found active in at least one of six cancer cell lines. Of all enhancers, we took 4,833 as (cell-line) specific enhancers, and 123 as universal enhancers. Specific enhancers each were active in only one cell line, whereas universal enhancers each were active in four, five, or six cell lines. An intermediate term, non-specific enhancer, was defined for enhancers active in 2~6 cell lines, and these non-specific enhancers totaled 1,134.

GTF (Gene Transfer file) file for the human genome GRCh38 was downloaded from Ensembl (https://uswest.ensembl.org) on 10/17/2018. We kept unique intervals for exons, and organized them into separate files for 47 distinct gene types.

Presumable binding segments of 26 human RNA-binding proteins were downloaded [18] and aligned to GRCh38 via the blastn application [39].

A total of ~2.61m eQTLs (expression quantitative trait loci) in the human genome GRCh37 were obtained from GTEx v7 (p<0.01, Fixed Effect model). 3.6% of these eQTL sites were deletions, for which we considered only the first nucleotides for simplicity.

Mutation Annotation Files for 33 cancer types (GRCh38) were downloaded from Genomic Data Commons (https://gdc.cancer.gov/), which maintains somatic mutation data generated by The Cancer Genomes Atlas (TCGA) project. Merging across all cancer types, we arrived at ~2.62m SNVs (Single-Nucleotide Variations) and ~165k indels.

From the ANNOVAR database [40], we downloaded the gnomAD datasets [19] which stored polymorphic variants among healthy populations. The GRCh38 dataset consisted of ~227k SNVs and ~66.4k indels.

From the UCSC Table browser, we downloaded computationally scanned Transcription Factor Binding Sites (TFBSs) located in GRCh37 [41], which included ~5.49m records.

From the ENCODE project, we downloaded six histone modification datasets that were assayed in human's muscle of trunk and were accessible in bigWig file formats. These six datasets featured H3K27ac, H3K27me3, H3K4me3, H3K9me3, H3K4me1, and H3K36me3, respectively.

Chromosome coordinates of potentially active LINE-1 (L1) transposons, including completely intact (FLI-L1s) and ORF2-intact (ORF2-L1s) LINE-1s, were downloaded from L1Base [42] (http://l1base.charite.de/l1base.php).

Database euL1db [43] and a published supplementary table [44] collected the confirmed insertion sites (target-site duplication) of human Retrotransposons Insertion Polymorphisms (RIPs). Because these two sources were largely non-overlapping, we merged them to form a single RIP dataset.

Low-interest variants (n = 167,144) recommended as a blacklist for human exome analysis (GRCh38) were obtained from a published supporting material [7].

A list of these miscellaneous genomic features, mapping to the identifiers in Fig 5, is provided in S3 Table.

## Miscellaneous statistical analyses

All statistical analyses, including the procedures encompassed in software package Polytrap, were conducted in R environment. We employed the hypergeometric probability model to test for the significant enrichment of MNRs/DNRs/TNRs in enhancers (Fig 2A) based on four key numbers: number of enhancers, number of enhancers embedding polytracts, number of nucleotides in reference genome, and number of nucleotides in reference genome occupied by polytracts. Confidence Interval (95%) of Relative Risk (Fig 2B) was calculated following the standard way [45]. In hierarchical clustering analysis (Fig 3B), distance metric used one minus correlation value and linkage choice was the average method. We employed the over-representation binomial probability model implemented in Polytrap to study RNA-editing events' enrichment in various polytract species, where the statistical significance was set at nominal p<0.01. Pearson correlation coefficient ($r$) was calculated and tested for $r \neq 0$ when we studied a possible linear correlation relationship between mean length and density of polytracts (S1 Fig).

## Supporting information

**S1 Table. Comparative summary statistics of polytracts between GRCh38 and GRCh37.**
*TNR species are designated with lowercase strings just to imply that the written TNR string actually accommodates multiple TNR motifs. For instance, TNR aat accommodates six repeat motifs: AAT, ATA, TAA, ATT, TAT, and TTA. This phenomenon is different from that of MNR and DNR, where the species name uniquely identifies a repeat motif.
(DOCX)

**S2 Table. Long DNRs and TNRs present in Polytract dataset yet absent in UCSC Microsatellites.** DNRs and TNRs of 15 or more units present in Polytract dataset yet absent in UCSC Microsatellites (GRCh38).
(TSV)

**S3 Table. Genomic features covered in polytract feature enrichment landscapes.**
(DOCX)

**S4 Table. Comparison of Polytrap against related tools.**
(DOCX)

**S1 Fig. Significant linear correlation exists between mean length and density of TNRs across chromosomes.**
(TIF)

**S2 Fig. Distribution of each polytract species across seven genomic region categories.** A, MNR. B, DNR. C, TNR.
(TIF)

**S3 Fig. Replicated TNR length distribution and concomitant TNR species clustering.** From the total 1,418,147 TNR tracts, we identified a subset of 39,937 long polytracts that each contained six or more (complete or incomplete) trimer units (i.e., length $\geq$ 18nt). By repeating the same precedent analytics as performed by Kozlowski et al., we revealed largely the same length distributions for the ten TNR species, and rendered a similar TNR clustering based on a matrix of pairwise Kolmogorov-Smirnov test statistics (S3 Fig). A, polytract length distribution dissected by TNR species. B, heatmap graph showing the Kolmogorov-Smirnov statistics D resulting from pairwise comparisons of polytract lengths among all 10 TNR subtypes.
(TIF)

**S4 Fig. Relative risk of cancer somatic mutation within polytracts versus whole genome.** TCGA somatic SNVs were significantly over-represented within TNR polytracts but not MNR/DNR polytracts (S4 Fig, A vs. B). Because TCGA mutations were heavily biased to exomes due to experiment design and TNR polytracts have a greater exonic portion than MNR/DNR, we suspected the unique enrichment of somatic SNVs within TNR might just be a trivial reflection of the elevated exonic component of trimer tracts. To interrogate this suspect, we restricted the enrichment analysis to non-coding SNVs only, finding the qualitative distinction between MNR/DNR and TNR persisted for nearly all cancers (S4C Fig). In conclusion, we observed significant enrichment of cancer SNVs within TNR but not within MNR/DNR, even after taking into account the disparate genomic compositions of polytract clades. A, enrichment of SNVs/indels within combined monomer/dimer tracts. B, enrichment of SNVs/indels within trimer tracts. C, enrichment of non-coding SNVs within monomer/dimer or trimer tracts. Asterisk (*) indicates significant enrichment of mutations within polytract regions ($p < 0.01$).
(TIF)

**S5 Fig. Locational under-representation of certain RNA-editing events in polytracts.** By setting the directionality option of Polytrap to "under," we switched to investigate the under-representation tendency of three classes of RNA-editing events in polytracts, and found A-to-G events were under-represented in almost all polytract species, excluding A/T, ACC, and hinges. S5 Fig is manifested in analogous format to Fig 4, but the significance asterisk notation (*) in panel A and color shading in panel B designate under-representation rather than over-representation. A, barplot denotes extremity of Relative Risk (RR), with significant (p<0.01) under-representation tendency annotated with asterisk (*). B, heatmap of decreased RR of polytract bearing an RNA-editing event, depicted for combinations between editing event classes and polytract species.
(TIF)

## Acknowledgments

We thank Luis Nassar and Matthew Speir from UCSC Genomics Institute for assistance in compilation of the tract data hub. We appreciate the four reviewers of our manuscript who raised important concerns and made constructive suggestions throughout rounds of revisions.

## Author Contributions

**Conceptualization:** David C. Samuels, Olufunmilola Oyebamiji, Yan Guo.

**Data curation:** Hui Yu, Shilin Zhao, David C. Samuels.

**Formal analysis:** Hui Yu, Quanhu Sheng, Olufunmilola Oyebamiji.

**Funding acquisition:** Yan Guo.

**Investigation:** Hui Yu, Shilin Zhao, Quanhu Sheng.

**Methodology:** Hui Yu, Shilin Zhao, Huining Kang.

**Project administration:** Scott Ness, Yan Guo.

**Resources:** Scott Ness, David C. Samuels, Ying-yong Zhao.

**Software:** Hui Yu.

**Visualization:** Olufunmilola Oyebamiji, Ying-yong Zhao.

**Writing – original draft:** Hui Yu.

**Writing – review & editing:** Hui Yu, Shilin Zhao, Scott Ness, Huining Kang, Quanhu Sheng, David C. Samuels, Olufunmilola Oyebamiji, Ying-yong Zhao, Yan Guo.

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
