## [Decision Letter · Decision Letter 0]

7 Jan 2020

Dear Dr Yu,

Thank you very much for submitting your manuscript 'Non-canonical RNA-DNA differences and other human genomic features are associated with very short tandem repeats' for review by PLOS Computational Biology. Your manuscript has been fully evaluated by the PLOS Computational Biology editorial team and in this case also by independent peer reviewers. The reviewers appreciated the attention to an important problem, but raised some substantial concerns about the manuscript as it currently stands. While your manuscript cannot be accepted in its present form, we are willing to consider a revised version in which the issues raised by the reviewers have been adequately addressed. We cannot, of course, promise publication at that time.

Sincerely,

Lilia M. Iakoucheva, Ph.D.

Associate Editor

PLOS Computational Biology

William Noble

Deputy Editor

PLOS Computational Biology

[LINK]

Reviewer's Responses to Questions

**Comments to the Authors:**

Reviewer #1: The authors of the reviewed article (PCOMPBIOL-D-19-01709) aimed to catalog and characterize the mono- di- and tri-nucleotide tracts present in the reference human genome and looked for the associations of the tracts with numerous genomic features, e.g., RNA editing sites. Although, as admitted by the authors, this type of study is not completely new, however previous studies usually were focused on a particular type of repeats or particular genomic regions and often were performed in pre- or early-genomic era. Therefore, systematic complex descriptive analysis of all types (1-3 nt) of microsatellites still could have been rationalizing the study. Additionally, the authors have extended the analysis to relatively short repeat tracts (minimal length of 6, 6, and 9 nt for mono- di- and tri-nucleotide tracts, respectively) that are extremely abundant in the genome and which analysis/inclusion could have been a real novelty of the study. The study is well justified.

Nevertheless, in my opinion, the aspect of descriptive characterization of the repeats is not fully explored and is not informatively presented. The manuscript has also several serious drawbacks that would have to be clarified/corrected before considering the manuscript for publication.

1. The redundant repeats should be merged as it is done for triplets. There is no need to distinguish complementary tracts (e.g., A and T or GA and TC) unless located and considered in the context of the direction of a gene or other genomic elements (it is not the case of the present version of the study). Otherwise, the genomic orientation of repeat tracts is random, for example, please see almost identical statistics for complementary tracts in Appendix 1. The distinguishing the complementary tracts makes a lot of analyses redundant and unnecessarily confusing.

2. The authors very inconsistently use names, descriptions for particular repeat tracts. The single/specific name should be selected for a particular tract at the beginning and then it should be used consequently throughout the entire manuscript, including pictures and supplementary materials. For example, in the Introduction the authors abbreviate trinucleotide repeats as TNRs but than use different terms, including tri-nucleotide tracts, Tri-nucleotide tracts, tri-tracts, trimers, tri, tri-, as well as again explain and use the abbreviation TNR. I would suggest to consequently to trinucleotide repeats (TNR) abbreviate mononucleotide and dinucleotide repeats (eg. MNR and DNR, respectively) and use them consequently in the manuscript.

3. Results, 2nd paragraph: “ We identified the same three most exome-prominent tri-tracts as previous researchers did (9): GAC, CGG, and CAG, but presenting them in a different exome-prominence order (GAC > CGG > 121 CAG, rather than CGG>CAG>GAC (9)).” In the previous study (9) the 3rd most exome prominent TNR was AGG (not GAC). GAC was the 3rd most exome overrepresented (see the difference between Fig. 1A and Fig. 1B in (9)). GAC was the least frequent TNR, with only 16 occurrences in the genome, including 3 in exons.

4. Results, 4th paragraph: It should be clarified whether density is a number of tracks per Mb or a number of nucleotides in tracks per Mb. It is critical for understanding the last sentence in the paragraph. To this paragraph but also generally to the study – due to the high overrepresentation of shortest tracts (extremally not normal distribution) the differences between tracks lengths are very small and therefore analyses/comparisons of means or even medians of the repeat lengths are of little value (e.g. differences in repeats lengths between chromosomes in Appendix 1.

5. Results, 5th paragraph, the part starting with “An intuitively plausible ...” to the end of the paragraph. I got lost in this part. Despite that I am quite familiar with the subject, it took me quite a long time to comprehend this paragraph. I would suggest to reconsider and to rewrite the paragraph.

6. Results: I do not understand the sentence ‘AAT is the most prevalent TNR species (Fig. 1E), bearing a potential replication regulatory role in partnership of 4′,6‐diamidine‐2‐phenylindole (DAPI) (15)’.

7. Results, the paragraph starting with line 196: As mentioned above, as differences in TNRs length are very small the analysis described in the paragraph is of little value. It would be more interesting and more informative to compare the frequency of TNRs occurrence in particular genomic/gene regions. It should be compared separately for each TNR type, and in genes also with consideration of genes orientation. Same for other types of repeats.

8. Resalts, the sentence starting with line 223: Please explain the better.

9. Results, line 197: Why one-tiled test? Please justify. I think it should be the two-tiled test.

10. It is not always clear to me what exactly was used as a background for calculation of expected frequency or cooccurrence of repeats with particular genomic features. It is important because some repeats may associate with genomic regions enriched with the tested feature and it may lead to a spurious association. For example, it is not clarified what type of cancer somatic mutations were extracted from TCGA (whole-genome?, total whole-exome including exon flanking sequences?, only exon sequences including UTRs?, or exon ORFs excluding UTR?) and in consequence what was used as a target and background sequence for analysis. It is known that particular repeat types associates with these subregions, therefore, some comparisons may lead to false-positive spurious associations,

11. Materials and Methods, ‘in human reference genomes GRCh37 and GRCh38, respectively’: What does it mean? In which assembly which repeats types have been analyzed? Why two assemblies were used?

12. Materials and Methods, ‘H1 (alternative hypothesis) - the concerned genomic feature is distributed in polytract territory at a frequency higher than random expectation.’ Why only frequency higher than expected were considered as H1? In my opinion, both over and underrepresentations should be considered and accordingly, two-sided tests should be used. Please explain.

13. Appendix 1 and corresponding text fragments: It would be more interesting to see comparisons of particular repeat types occurrence in particular genome/gene subregions (5’UTR, 3’UTR, ORF, introns, intergenic regions) instead of in different chromosomes, in which occurrence mostly depends on chromosome length and therefore frequency or density do not differ substantially. The exception may be chromosome Y, X, and MT. Autosomes may be combined.

14. The legend in Figure 1B: What is it ‘exome splicing’? What about the coding sequence subregion? It is not indicated.

15. Figure 3B and corresponding fragment of Results: Heatmap and dendrogram do not show unique distributions of CGG, ACC, and ATT. ATT show very similar distribution with AGG and quite similar distribution (same clade) with all other but CGG and ACC TNRs. The interpretation of Fig 3B should be reconsidered. Also, there is something wrong with the bitmap. It shows that the comparison of AAT with AAT (with itself) gives D-value different from 0. The same is for AGG. The same problem is with the bitmap in the supplementary Figure (Appendix 6).

16. Figure 4A and corresponding fragments of Results. It is shown that RNA editing events are not only enriched but also underrepresented in some repeat types (eg. A2G is underrepresented in dinucleotide tracts and TNRs). The fold of these changes is similar to some positive changes but the significance of these changes is not indicated. The decreases in RNA-editing events are not commented in the text. Please explain/comment on the lack of significance of the negative changes and the fact that repeats generate both increases and decreases of RNA-editing events or artifacts.

17. Figure 5: It is very surprising that the significant associations ordered according to p-values correlate perfectly between repeat classes, i.e., the same order of genomic features in mono- di, and tri-nucleotide repeats. It may suggest spurious associations. I would suggest double-checking.

18. Figure 5: It would be interesting to see also fold changes (not only p-values) for particular features. To see whether an association is positive or negative. Also, it would be interesting to see how the particular subtypes of mono-, di-, and tri-nucleotide tracts associate with the genomic features.

19. The list of identified tracts, their type, genomic coordinates, and potentially associated genomic features should be always available for potentially interested readers. The UCSC custom tracts for the particular repeats are deposited on private google drive. I have bad experiences with “upon request” or private repositories. They usually do not work shortly after publication. Also, there is no need to mention the custom tracts for other organisms in the abstract if they are not analyzed or discussed in the paper.

20. Appendix 1: The title of the subtable (spreadsheet) ‘genomeOccupancy’ should be rather “chromosomeOccupancy”.

Reviewer #2: This paper presents an updated census of very short tandem repeats (polytracts) particularly in human reference genome GRCh38. Further, authors report their findings of enrichment and association analysis of polytracts (mainly monomers, dimers and trimers) and RNA-edit events both canonical and non-canonical with respect to various genomic features. The authors also provide polytrap, a software tool to perform such association analysis.

Overall, these findings could be of interest to a broad audience in the scientific community. However, it would be a bit of a stretch to confirm such findings when the methodology is not well presented. I would highly recommended the authors to rewrite their materials and methods to explicitly present their approach and how they conducted their analysis. In the following, I indicate a few points that should be addressed:

• There are fundamentally major weaknesses in this manuscript:

o Authors claim that previous analysis of polytracts are outdated and imprecise. I find the argument of improved sequencing technology and better curation of the current version GRCh38 reference genome more plausible, but beside that, the authors fail to present evidence of how their analysis has led to improved detection and characterization.

o Authors do not provide the theoretical framework behind the identification, detection and categorization of polytracts. Authors relied on BSgenome, a Bioconductor package. I find the details in Methods and Materials insufficient in terms of polytracts detection and identification. Authors must include the underlying logical framework and methodology used to infer polytracts from a DNA sequence (reference genome). Authors should indicate whether the method considered leads to improved detection of polytracts especially monomers and dimers that have not been widely reported.

o The paper as it is now has no details or description of the statistical analysis for the results presented. Authors should include a section describing and discussing the statistical analysis and statistical tests performed.

o Authors present polytrap and yet have not described its algorithmic idea and how it works. I encourage the authors to include such information as well.

• The Authors use the binomial distribution and clearly express the hypotheses they wish to test. The authors should explain why they considered this distribution and any underlying assumptions made and how this relates to the downstream association analysis.

o How do you define and determine the tract territory?

o L387-L392: The singleton mode and the multiplex mode, it should be helpful to visually illustrate the different modes if possible.

o Authors use terms like tract number, density, tract volume and genome occupancy and yet do not explain them in the main text nor in the supplementary materials [Supp. S1].

• Location of miscellaneous genomic features

o Authors should briefly discuss why they considered the specified genomic features and discuss briefly their biological importance for instance with respect to RDDs

o It appears to me that the authors only listed materials gathered for their analysis. They fail to present how they actually conducted and performed their analysis. You have collected data from various sources with potentially different genomic coordinates (GRCh37 vs GRCh38, for instance) and yet you do not mention how all that information was integrated and used.

o Overall, the materials and method part of the paper needed to be significantly improved

• Minor issues:

o Provide citation L151

o L215: For consistency, authors should report their RR as done at L223 also to avoid confusion with citation notations.

o L323-L329: I would encourage authors to discuss and put forth the argument of why analysis of A2G editing events are needed in the context of host polytract and probably mention their current manuscript in preparation rather than reporting their findings.

o Proofread the text

o The figures could be a of a better quality and higher resolution

Reviewer #3: This article is well written and thoroughly tries to review/census monomers, dimers, and trinucleotide (tandem) repeats found in GRCh38, by correlating their distribution and composition to specific genomic features and loci.

However, as a first observation, no comparison is made with, for example, simple repeats (UCSC track, last updated on March 2019) and works associating them with human diseases and other genomic features, as well as with the mentioned 700k catalog by Willems et al. and other catalogues and censuses that can be easily retrieved from online (e.g. https://www.frontiersin.org/articles/10.3389/fgene.2018.00155/full).

More importantly, no accurate details and definitions are given for the discovery of such (tandem) repeats, while these could clarify some of the statistical findings and help identifying potential biases with the analysis; sophisticated software and methodologies are nevertheless available from other works, which allow for a flexible and robust tandem repeat finding (e.g. TRF and more recent tools). Reverse complement is also only considered for trinucleotides, while the analysis should be comparable between all three types for some statistics.

Furthermore, the fact that the main differences between DNA and RNA are found inside or adjacent to monomer repeats might easily suggest that these come either from sequencing errors, alignment consensus errors, or possible (low-frequency) polymorphisms. Indeed, monomers, especially long monomers, can be hardly sequenced well and are known to create several artefacts and analysis issues both in sequencing, pre-processing, and post-processing phases, as well as are more prone to indels of a different nature (than the repeat itself) than the other two types. Not surprisingly, significant differences in the distribution of somatic mutations in cancer genomes is noted for both them and trinucleotides (i.e. the size of codons). I would thus suggest to avoid, at least for the moment, mainly focusing on these discoveries both in the title and the abstract.

Finally, no information is available on the created software, as well as on its memory footprint and general performance.

Reviewer #4: Based on the new human reference genome GRCh38, this manuscript studied the genome-wide short tandem repeats (STR) characteristics, in particular the monomer, dimer, and TNR and the implication on RNA-DNA differences and other genome features, which has provided new insight into our understanding of the impact of STR on genome structure and function. In addition, they developed a software to characterize STRs in the genomes, which will be helpful for other researchers. However, some questions are not reasonable and needed to be clarified.

Major comments:

1.The authors used infrastructure packages from Bioconductor (www.bioconductor.org;

DOI: 10.18129/B9.bioc.BSgenome) to identify monomer, dimer, and TNR in human reference genomes，and found the abundance of very short tandem repeats in the new reference genome was much higher than those in the previous reference genome. I am wondering why the authors did not apply some traditional methods such as RepeatSeq or lobSTR to recognize STRs to the reference genome? The authors should justify their methodology and figure out whether the new version of reference genome or the data procession cause such differences on STR abundance. The authors also can check this literature to find better way to deal with STRs.

Willems T, Zielinski D, Yuan J, Gordon A, Gymrek M, Erlich. Genome-wide profiling of heritable and de novo STR variations. Nature Methods. 2017, 14, 590–592.

2.Considering a large number of previous studies on human genome STRs and their characteristics, it is suggested that the author supplement the differences between GRCH38 and previous version, and add the comparison of their results with other studies based on the old version of reference genome.

3.Although the detail analysis has been conducted on the GRCH38 reference genome in the present study, it is only based on one genome. As known for us, short tandem repeat sequences show high polymorphism between genome and genome. How can an analysis based on a single genome reflect the real characteristics of STRs in human or how high is the reliability? The authors should clarify.

4.Based on the new reference genome, the authors analyzed the RNA editing sites associated with STRs, which is one of the highlights of this manuscript. It is suggested that the authors should strengthen this in the Results, in particular they should add some comments in the Discussion to highlight the significance of STRs on RNA editing events in combination with relevant literatures on human RNA editing in recent years.

5.The manuscript developed a software to assist with custom association analysis of polytracts with respect to any localizable genomic feature. Unfortunately, trying to assist the Polytrap is failed to me. Furthermore, the link provided by the manuscript about genomic coordinates of polytracts in nine model organisms is also unavailable. Please provide validated way for using and evaluating the Polytrap. And i would like to know if the Polytrap could be used for population scale analysis.

6.There are many recent literatures about STRs or RNA editing in human genome. In the discussion part, the authors seldom used the literatures and compared their results with the previous studies. Many sentences in the discussion are repeat the results, thus suggest the authors to improve.

Minor comments:

In the Abstract and Keywords, the authors used C-to-U RNA-editing, however in the text part, they used A2T or C2T. This should be revised.

**Have all data underlying the figures and results presented in the manuscript been provided?**

Reviewer #1: No: The lists of identified mono-, di-, and tri-nucleotide tracts provided as custom track files for UCSC Genome Browser are deposited (linked) only on private Google Drive. These lists are very important result of the study and should be deposited in public database or if possible as supplementary materials.

Reviewer #2: Yes

Reviewer #3: Yes

Reviewer #4: No: the polytrap is unaccessable

PLOS authors have the option to publish the peer review history of their article (what does this mean?). If published, this will include your full peer review and any attached files.

Reviewer #1: No

Reviewer #2: No

Reviewer #3: No

Reviewer #4: No

---

## [Decision Letter · Decision Letter 1]

27 Mar 2020

Dear Dr. Yu,

Thank you very much for submitting your manuscript "Non-canonical RNA-DNA differences and other human genomic features are enriched within very short tandem repeats" for consideration at PLOS Computational Biology. As with all papers reviewed by the journal, your manuscript was reviewed by members of the editorial board and by several independent reviewers. The reviewers appreciated the attention to an important topic. Based on the reviews, we are likely to accept this manuscript for publication, providing that you modify the manuscript according to the review recommendations.

Sincerely,

Lilia M. Iakoucheva, Ph.D.

Associate Editor

PLOS Computational Biology

William Noble

Deputy Editor

PLOS Computational Biology

[LINK]

Reviewer's Responses to Questions

**Comments to the Authors:**

Reviewer #1: Detailed comments:

1. L149-L150: The high GC content of hinges connecting AT-reach tracts (i.e., A/T_A/T or TA_TA) is not surprising. I would say that it is rather expected, as the presence of either A or T in the hinge positions would, in most cases, extended the tracts and would not be counted. The same I think applies to GC-reach tracts that, I expect, are mostly separated by AT-reach hinges.

2. Table S2: I do not know whether I understand correctly the column ‘Polytract As Microsatellite’. I would expect that the values in the column should be equal (should not be higher) with the values in the ‘Microsatellite’ column. It is the case for most but not for all polytracts. Please explain or correct it.

3. Figure 1E and Figure S2: What is up_down_stream? How it is defined?

4. Maybe I am too pedantic but why the authors sometimes use a lower-case and sometime upper-case letter for different polytracts (e.g., TA or ta) or why sometimes use for DNR e.g., CT and in other places CT/GA. For readers, especially when someone browses the article quickly such inconsistencies may be confusing.

5. Figure S2 and corresponding text: It is worth to note that GC DNRs are strongly enriched in 5’UTR, exonic, and up_down_stream regions.

6. Does ‘exonic’ mean protein-coding part of exons or exons including UTRs?.

7. According to Figure 3B (bitmap), CGG TNRs are more similar to AATs that to AACs. The fragment of text commenting Figure 3 should be reconsidered.

8. Fragment encompassing L189~L191: Figure 3A does not show repeats “6U or more” but ‘5U or more’ (15+).

Reviewer #2: The authors of the revised manuscript “Non-canonical RNA-DNA differences and other human genomic features are enriched within very short tandem repeats” have made substantial improvement over the original manuscript. They have satisfactorily addressed most of the concerns and questions I raised earlier, but I still find the article missing an important component: description of performed statistical analyses. Authors performed various statistical tests as illustrated throughout the manuscript mainly in the Results section and shown in the Figures. Though the fundamental statistical analysis incorporated in the software polytrap is well described in the Materials and methods, a detailed description of the performed ad hoc statistical analyses is missing [including which statistical softwares were used]. In addition, it would be necessary to know for instance based on polytrap output, what is the cut off p-value for statistical significance in enrichment analysis.

In addition, the figures especially Figures in the main manuscript are not of publishable quality [when you print them you can hardly read the figure annotations and labels]. In addition, in Figure 6, N is the total number of nucleotides in the genome, but it is indicated as if it is the total number of nucleotides excluding the total number of nucleotides occupied by polytracts. By your definition, N in Fig. 6 should be illustrated by blue color + grey color.

Minor typos on lines L412 [I suppose you meant hypergeometric model instead of hypegenometric model], 442-443 [the word “instead” makes the sentence reads a little bit contradictory to the previous sentence].

Reviewer #3: The article has been reviewed fairly well and most issues have been addressed.

Although an exact tandem repeat finder may hide useful approximate tandem repeats or shorten/break up longer chains, and therefore somehow mislead the statistics you have found, your catalog could help reporting new locations/repeats (absurdly) overlooked before (e.g. in UCSC Simple/Microsatellite Repeats). In this regard, have you checked that your new locations/repeats are correct? Could you justify why they don't appear in the UCSC catalog? Can you confirm that the new locations/repeats are not present in catalogs like the 700k by Willems et al.?

In addition, I would strongly suggest to emphasize earlier, rather than in the Discussion section, that the main differences between DNA and RNA found within or adjacent to monomer repeats may have been caused by artifacts (e.g. sequencing errors, alignment consent errors) or potential (low-frequency) polymorphisms, and similarly for what is reported for somatic mutations in cancer genomes for both mononucleotides and trinucleotides (i.e. the size of codons).

Reviewer #4: The authors have almost answered questions concerned by me. They added related information in methods part, and rewrited the disscussion. The new version looks much better than last version. I suggest to accept the manuscript to publish on the Plos Computaional Biology.

**Have all data underlying the figures and results presented in the manuscript been provided?**

Reviewer #1: Yes

Reviewer #2: Yes

Reviewer #3: Yes

Reviewer #4: Yes

PLOS authors have the option to publish the peer review history of their article (what does this mean?). If published, this will include your full peer review and any attached files.

Reviewer #1: No

Reviewer #2: No

Reviewer #3: Yes: Davide Verzotto

Reviewer #4: No
---

## [Decision Letter · Decision Letter 2]

19 May 2020

Dear Dr. Yu,

We are pleased to inform you that your manuscript 'Non-canonical RNA-DNA differences and other human genomic features are enriched within very short tandem repeats' has been provisionally accepted for publication in PLOS Computational Biology.

Best regards,

Lilia M. Iakoucheva, Ph.D.

Associate Editor

PLOS Computational Biology

William Noble

Deputy Editor

PLOS Computational Biology

Reviewer's Responses to Questions

**Comments to the Authors:**

Reviewer #1: No more comments.

Reviewer #2: The authors have made satisfactorily and significant improvements compared to the last manuscript.

**Have all data underlying the figures and results presented in the manuscript been provided?**

Reviewer #1: Yes

Reviewer #2: Yes

PLOS authors have the option to publish the peer review history of their article (what does this mean?). If published, this will include your full peer review and any attached files.

Reviewer #1: Yes: Piotr Kozlowski, Institute of Bioorganic Chemistry, Polish Academy of Sciences, Poznan, Poland

Reviewer #2: No

---

## [Editor Report · Acceptance letter]

2 Jun 2020

PCOMPBIOL-D-19-01709R2 

Non-canonical RNA-DNA differences and other human genomic features are enriched within very short tandem repeats

Dear Dr Yu,

I am pleased to inform you that your manuscript has been formally accepted for publication in PLOS Computational Biology. Your manuscript is now with our production department and you will be notified of the publication date in due course.

With kind regards,

Matt Lyles
